# Treatment outcomes and prognostic factors in external auditory canal squamous cell carcinoma

**Byung Chul Kang**[1], **Se Eun Yi**[2], **Woo Seok Kang**[2], **Joong Ho Ahn**[2], **Hong Ju Park**[2], **Jong Woo Chung**[2]*

1 Department of Otorhinolaryngology-Head and Neck Surgery, Ulsan University Hospital, University of Ulsan College of Medicine, Ulsan, Korea, 2 Department of Otorhinolaryngology-Head and Neck Surgery, Asan Medical Center, University of Ulsan College of Medicine, Seoul, Korea

* jwchung@amc.seoul.kr

## Abstract

External auditory canal carcinoma (EACC) is a rare and aggressive malignancy with substantial variability in prognosis depending on tumor stage and adjacent structure involvement. We retrospectively reviewed 56 patients with histologically confirmed squamous cell carcinoma of the external auditory canal treated at a tertiary referral center between 2000 and 2022. Clinical data including demographics, tumor stage, treatment modalities, surgical approach, and survival outcomes were analyzed. Kaplan–Meier survival curves and Cox proportional hazards regression were used to identify prognostic factors for overall survival (OS) and disease-specific survival (DSS). Of the 56 patients (mean age 61.6 years; 46.4% female), 30 had early-stage (T1–T2) and 26 had advanced-stage (T3–T4) tumors. The 5-year OS rates were 100.0% for early-stage, 60.0% for T3, and 42.0% for T4 disease. Advanced T-stage, nodal metastasis, and abutment to vascular structures such as the carotid artery or jugular bulb were significantly associated with worse outcomes. In multivariate analysis, younger age, vascular abutment, and nodal metastasis were independent negative prognostic factors. En bloc resection with clear margins was associated with improved survival. These findings emphasize the importance of early diagnosis and meticulous surgical planning to achieve complete resection and optimize outcomes in patients with EACC.

## Introduction

External auditory canal carcinoma (EACC) is a rare malignancy arising from the epithelial lining of the external auditory canal (EAC), accounting for fewer than 0.2% of all head and neck cancers [1]. Despite its low incidence, EACC presents significant management challenges due to its aggressive nature and close proximity to critical structures, including the facial nerve, temporal bone, dura mater, and major blood vessels [2]. Clinical outcomes are closely associated with tumor stage at diagnosis, with early-stage disease often being curable with surgical resection, whereas

**Data availability statement:** The anonymized dataset underlying the findings described in this manuscript has been uploaded as Supporting Information (S1_Dataset_EACC. xlsx).

**Funding:** The author(s) received no specific funding for this work.

advanced-stage disease is associated with poor prognosis and high morbidity [3]. Surgical resection remains the cornerstone of treatment, often combined with adjuvant radiotherapy (RT) or chemotherapy in advanced cases. The Modified Pittsburgh Staging System is widely used to guide treatment decisions [4].

Achieving complete tumor removal can be technically challenging, particularly with extensive bony invasion [3]. Positive surgical margins and perineural invasion have been linked to recurrence and poor survival [1,5–7], while facial nerve involvement, lymph node metastasis, and distant spread also worsen prognosis [2,4,8]. Adjuvant RT is commonly used in cases with high-risk features such as positive margins, perineural invasion, or regional metastasis [2,9,10]. However, its survival benefit remains uncertain [1]. Chemotherapy is generally reserved for unresectable or metastatic cases, and its optimal role in neoadjuvant or definitive settings remains unclear [1,11]. Despite these multimodal approaches, outcomes for advanced EACC remain poor, with 5-year survival rates ranging from 20% to 40% [3]. These findings underscore the need for reliable prognostic indicators and optimized treatment strategies to improve survival.

Squamous cell carcinoma (SCC) of the external auditory canal (EAC), the most common histologic subtype of EACC, is characterized by local invasiveness and a high propensity for nodal metastasis driven by epithelial–mesenchymal transition and overexpression of matrix metalloproteinases [12–14]. Consequently, SCC of the EAC is often diagnosed at an advanced stage (cT3–T4 in 57.1% of cases), and is associated with poor overall survival (OS) and disease-specific survival (DSS) [15], whereas basal cell carcinoma (BCC), demonstrates more indolent behavior [16]. Given the aggressive clinical course and anatomical complexity of EAC SCC, this study aimed to evaluate treatment outcomes and identify key prognostic factors postoperative survival in a cohort treated over two decades at a single tertiary referral center. By analyzing detailed anatomical invasion patterns and surgical strategies, we sought to refine prognostic assessment and inform future management of this rare but aggressive malignancy.

## Materials and methods

This retrospective study included patients diagnosed with SCC of the EAC who underwent surgical treatment between 2000 and 2022 at a single tertiary referral center. A total of 56 patients met the inclusion criteria. Data were obtained from electronic medical records. The data were accessed for research purposes between 07/05/2024 and 01/07/2025. Institutional Review Board approval was obtained before study initiation (2024−0596), and the requirement for informed consent was waived due to the retrospective design.

### Patient selection

The inclusion criteria were as follows: (1) histologically confirmed SCC of the EAC, (2) availability of complete medical records and follow-up data, and (3) surgical treatment performed with curative intent. Patients were excluded if they had non-SCC histologies, incomplete medical records, or received palliative treatment.

## Data collection

Data collected included demographic variables (age, sex), clinical presentation, tumor characteristics (stage, size, location), treatment modalities, surgical margins, and postoperative outcomes. Age was analyzed as a continuous variable in both univariate and multivariate Cox regression models; therefore, no dichotomized age cutoff was used for comparison between "young" and "old" patients. Follow-up data were used to evaluate recurrence, complications, and survival outcomes. Tumors were staged according to the Modified Pittsburgh classification system, which is widely used for the assessment of EACC.

Initial surgical planning was based on clinical staging according to preoperative imaging and physical examination using the modified Pittsburgh classification system. Final tumor staging used for analysis in this study was determined by postoperative pathological assessment (pT) according to the same staging system.

## Surgical procedures

All patients underwent surgical resection tailored to tumor stage and extent. Surgical approaches ranged from sleeve resection and lateral temporal bone resection (LTBR) for early-stage disease to lateral or subtotal temporal bone resection (STBR) for advanced cases. Neck dissection and parotid management were performed according to tumor stage and the planned extent of temporal bone resection. In T1 cases treated with sleeve resection, neither neck dissection nor parotidectomy was performed. For T1–T2 cases undergoing LTBR, superficial parotidectomy and selective neck dissection (levels I–II) were routinely performed, and no parotid or lymph node metastasis was detected in this early-stage group. In patients with T3–T4 disease, total parotidectomy was performed in conjunction with STBR, and neck dissection included levels I–III, with level IV added when preoperative imaging suggested nodal involvement. Advanced cases treated with extended LTBR underwent superficial parotidectomy. The surgical objective was complete tumor resection with clear surgical margins while preserving critical anatomical structures where feasible. All resections were planned and initiated as en bloc resections. Complete en bloc resection was achievable for all T1–T3 tumors. In T4 tumors, where complete en bloc resection of all involved adjacent structures was anatomically unfeasible, en bloc temporal bone resection was performed first, followed by additional targeted removal of residual involved tissues, including cases with dural involvement, carotid artery abutment, jugular bulb abutment, or Eustachian tube involvement. Postoperative adjuvant therapy was planned according to institutional treatment principles using a risk-adapted approach for patients with advanced disease or high-risk pathological features, including positive or close surgical margins, perineural invasion, or nodal metastasis.

## Statistical analysis

Descriptive statistics were used to summarize demographic and clinical characteristics. OS rates were calculated using the Kaplan–Meier method. Kaplan–Meier survival curves were generated using Python with the Matplotlib library. Cox proportional hazards regression was performed using R software (version 4.4.1), with univariate and multivariate analyses conducted to identify prognostic factors associated with recurrence and survival. Age was analyzed as a continuous variable and not dichotomized into categorical groups. Variables with a p-value < 0.10 in univariate analysis, as well as clinically relevant factors, were entered into the multivariate Cox proportional hazards model. Variables analyzed included tumor stage, surgical margins, anatomical involvement, and cervical lymph node metastasis. Statistical significance was defined as $p < 0.05$. All analyses were conducted using the survival and survminer packages in RStudio (Posit software, PBC).

## Results

A total of 56 patients diagnosed with EACC were retrospectively analyzed. The cohort comprised 36 male and 20 female patients, with a mean age of 61.6 years. **Table 1** summarizes the distribution of patients by T stage, sex, and mean age.

**Table 1. Distribution of patients by age, sex, and T stage.**

|          | T1   | T2   | T3   | T4   | Total | Mean age |
|----------|------|------|------|------|-------|----------|
| Male     | 10   | 8    | 2    | 16   | 36    | 62.9     |
| Female   | 8    | 4    | 3    | 5    | 20    | 60.8     |
| Total    | 18   | 12   | 5    | 21   | 56    | 61.6     |
| Mean age | 60.1 | 66.5 | 64.8 | 59.3 |       |          |

Mean age values are presented in years. "T" indicates tumor stage according to the TNM classification. TNM stages represent final pathological staging (pTNM).

## Patient distribution by T stage and demographics

Tumor staging was performed using the Modified Pittsburgh classification system. Among the 56 patients, 18 were classified as T1, 12 as T2, 5 as T3, and 21 as T4. T1 and T2 stages had a relatively even sex distribution, whereas T4 stage was more prevalent in male patients (n = 16) compared with females (n = 5). Two patients who were initially staged clinically as T2 were downstaged to pathological T1 after histopathological examination due to the absence of true bone erosion. The mean age differed by stage, with T2 patients having the highest mean age (66.5 years) and T4 patients the lowest (59.3 years). Female patients had a slightly higher mean age (62.9 years) than male patients (60.8 years).

## Surgical methods and survival outcomes

Surgical interventions included sleeve resection, LTBR, STBR, and extended LTBR. Sleeve resection was performed exclusively in four patients with T1 lesions. No T2 or more advanced lesions were treated with sleeve resection; all T2 cases underwent LTBR. LTBR was the most common procedure for early-stage disease, used in 14 T1 and 11 T2 lesions. For T1 and T2 lesions, no patients received adjuvant RT. All T1 and T2 tumors achieved negative surgical margins on final pathological assessment. During the entire follow-up period, none of the patients with T1 or T2 disease developed local or regional recurrence or distant metastasis. STBR was the predominant surgical approach for advanced-stage disease, and extended LTBR was performed in two T3 and three T4 cases. All surgical procedures were planned with the intent of en bloc tumor excision. Complete en bloc resection was achieved in all T1–T3 cases. In T4 tumors, initial en bloc removal was attempted; however, when the lesion involved critical adjacent structures including the dura mater, carotid artery abutment, jugular bulb abutment, and Eustachian tube, additional stepwise resection was performed following en bloc excision to remove residual involved tissues and maximize oncologic clearance. Among the 19 patients with T4 disease, negative margins on the initial en bloc resection specimen were achieved in 5 cases, whereas positive margins were reported in 14 cases. Positive margins were mainly observed when tumors abutted critical adjacent structures, particularly the carotid artery, jugular bulb, or tympanic cavity. In these situations, following en bloc resection, additional targeted removal of residual involved tissues was undertaken. Intraoperative frozen-section analysis was used to confirm negative margins, and resection was continued until tumor-free margins were achieved at all accessible sites. For areas adjacent to the carotid artery or jugular bulb, where histologic sampling was not feasible, macroscopic inspection was performed. No cases demonstrated true intraluminal invasion of the carotid artery or jugular bulb. No parotid gland involvement or cervical lymph node metastasis was identified in T1–T2 cases, whereas nodal metastasis in T3–T4 cases was limited to clinically or radiologically suspicious nodes addressed during neck dissection. Most patients with T3 and T4 disease received postoperative adjuvant RT or CRT according to institutional treatment principles. Because the extent and pattern of adjacent structure invasion (such as the TMJ, mastoid, dura, and carotid artery) differ substantially by T stage, detailed comparison of anatomical invasion sites was limited to T4 lesions. Table 2 summarizes the surgical methods and survival outcomes of this advanced-stage cohort according to anatomical involvement. Of the 21 patients initially classified as T4, two were excluded from detailed anatomical-risk analysis because they died from causes unrelated to EAC SCC (acute

Table 2. Comparison of survival outcomes based on involvement areas and the type of surgical intervention.

| No | Stage | | | Survival | | Anatomical involvement areas | | | | | | | | | | Preop facial palsy | Surgical intervention |
|---|---|---|---|---|---|---|---|---|---|---|---|---|---|---|---|---|---|
| | T | N | M | Status | Months | Ant | MT | JB | PDL | ET | CA | PT | TMJ | TT | JF | | |
| 1 | 4 | 0 | 0 | X | 3 | – | – | △ | – | △ | △ | – | △ | △ | – | 4 | STBR |
| 2 | 4 | 2b | 0 | X | 6 | O | O | – | O | △ | – | – | – | – | – | 2 | STBR |
| 3 | 4 | 0 | 0 | X | 6 | O | O | – | – | O | △ | – | △ | △ | – | 1 | STBR |
| 4 | 4 | 1 | 0 | X | 6 | O | O | △ | O | – | – | – | – | △ | – | 1 | STBR |
| 5 | 4 | 1 | 0 | X | 8 | O | O | – | – | △ | △ | – | O | △ | – | 1 | STBR |
| 6 | 4 | 1 | 0 | X | 16 | O | O | △ | – | △ | – | – | △ | – | – | 2 | STBR |
| 7 | 4 | 0 | 0 | X | 23 | O | O | △ | – | – | – | – | – | – | – | 1 | Extended LTBR |
| 8 | 4 | 0 | 0 | X | 24 | – | O | – | O | – | – | – | – | – | – | 1 | Extended LTBR* |
| 9 | 4 | 2a | 0 | X | 31 | – | – | – | – | O | – | △ | – | – | △ | 1 | STBR |
| 10 | 4 | 0 | 0 | X | 36 | – | – | – | – | O | △ | △ | – | – | – | 1 | STBR |
| 11 | 4 | 0 | 0 | X | 37 | O | O | – | – | △ | △ | – | O | – | – | 5 | STBR |
| 12 | 4 | 1 | 0 | X | 50 | O | O | △ | – | – | – | – | – | – | – | 1 | STBR |
| 13 | 4 | 0 | 1 | X | 50 | O | – | – | O | – | – | – | △ | – | – | 1 | Extended LTBR |
| 14 | 4 | 0 | 0 | O | 218 | O | O | – | – | O | – | – | – | – | – | 3 | STBR* |
| 15 | 4 | 0 | 0 | O | 125 | – | O | – | – | – | – | – | – | – | – | 1 | STBR* |
| 16 | 4 | 0 | 0 | O | 76 | O | O | – | O | – | – | – | – | O | – | 1 | STBR* |
| 17 | 4 | 0 | 0 | O | 75 | O | O | O | △ | – | – | – | O | △ | – | 1 | STBR |
| 18 | 4 | 0 | 0 | O | 69 | – | – | – | – | O | – | O | △ | – | – | 1 | STBR |
| 19 | 4 | 0 | 0 | O | 37 | – | O | – | O | – | – | O | – | – | O | 1 | STBR* |

This table lists all 19 patients with T4 external auditory canal carcinoma (EACC), detailing their TNM stage, survival status ("X" indicates death; "O" indicates alive) and duration (in months), involved anatomical sites, preoperative facial nerve status (House–Brackmann grade), and type of surgical intervention.

In the anatomical involvement columns, "O" indicates tumor involvement with negative surgical margin on the initial en bloc specimen, "△" indicates tumor involvement associated with positive surgical margin on the initial en bloc specimen requiring additional stepwise excision for clearance, and "–" indicates no involvement.

An asterisk (*) identifies cases in which the initial en bloc resection specimen achieved overall negative surgical margins (initial R0 status).

Ant, Antrum; MT, Mastoid; JB, jugular bulb; PDL, parotid gland deep lobe; ET, Eustachian tube bony canal; CA, carotid artery; PT, petrous bone anterior to carotid artery; TMJ, temporomandibular joint; TT, tegmen tympani; JF, jugular fossa; STBR, subtotal temporal bone resection; LTBR, lateral temporal bone resection.

pneumonia at 29 months and an unidentified medical event at 41 months). Therefore, 19 patients were included in the T4-specific analysis presented in Table 2.

## Adjuvant therapeutic patterns and results

Postoperative adjuvant radiotherapy (RT) or chemoradiotherapy (CRT) was administered based on institutional treatment principles using a risk-adapted approach in cases with high-risk features or advanced disease. T1–T2 tumors achieved adequate surgical margins and had negative selective neck dissection (levels I–II); therefore, no adjuvant therapy was performed. Among patients with T3–T4 disease, postoperative RT was administered in 24 cases (total dose 5,040–6,660 cGy over 7–8 weeks), and chemotherapy was given to 21 cases, predominantly using cisplatin-based regimens. Most patients receiving chemotherapy were treated concurrently with RT. Among the 26 patients with T3–T4 disease, two did not receive postoperative RT and two did not receive chemotherapy because they declined treatment despite medical recommendation. Due to the heterogeneity of chemotherapy regimens over the 20-year study period and the fact that

adjuvant systemic therapy was often administered at patients' local hospitals following surgery at our institution, direct comparison of systemic therapy protocols was not feasible.

## Survival analysis

Patients with T1 and T2 tumors demonstrated excellent outcomes, with an OS rate of 100.0%; Kaplan–Meier survival analyses were therefore not performed for these groups. The mean follow-up duration was 86.2 months (range: 24–193 months) for T1 and 85.9 months (range: 32–151 months) for T2 patients. In contrast, patients with T3 and T4 disease showed more variable outcomes. The 5-year OS rate was 100% for early-stage (T1–T2), decreasing to 60.0% for T3, and 42.0% for T4 disease. Advanced T-stage was significantly associated with poorer survival ($p<0.01$). Kaplan–Meier survival curves are presented in **Fig 1**, illustrating lower survival probabilities in T4 compared with T3 patients. Mean follow-up durations for T3 and T4 groups were 21.0 months (range: 9–31 months) and 46.0 months (range 3–218 months), respectively.

Kaplan–Meier overall survival (OS) for the entire cohort, including T1–T4 disease. Number-at-risk values are shown below the plot, and censoring points are indicated on the curves.

In a univariate Cox regression analysis of patients with advanced T4 disease (n = 19), increasing age showed a non-significant trend toward improved survival (hazard ratio [HR] = 0.96, 95% CI: 0.92–1.00, $p=0.079$). Variables associated with significantly poorer prognosis included male sex (HR = 4.54, $p=0.023$), positive N stage (HR = 3.57, $p=0.028$), and carotid artery abutment (HR = 3.67, $p=0.033$).

In the multivariate Cox model, age remained a statistically significant independent protective factor (HR = 0.90, 95% CI: 0.83–0.98, p = 0.019). Carotid artery abutment (HR = 16.41, $p=0.050$), jugular bulb abutment (HR = 9.01, $p=0.067$), and N stage (HR = 4.40, $p=0.096$) showed borderline significance, indicating a trend toward increased risk of mortality. Other variables, including sex, Eustachian tube invasion, facial palsy, and tegmen tympani invasion, were not significantly associated with OS.

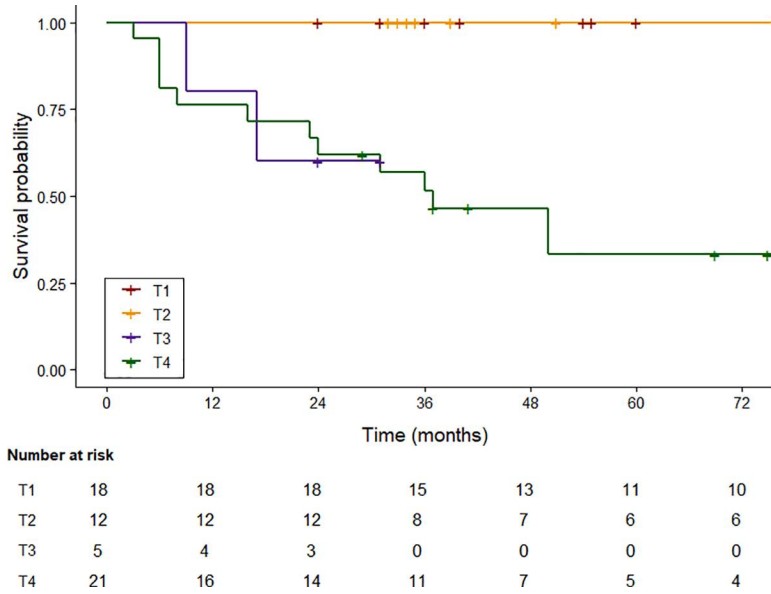

**Fig 1. Overall survival of patients with external auditory canal SCC (T1-T4).**

The multivariate model demonstrated high predictive accuracy, with a concordance index of 0.873. It was statistically significant based on the likelihood ratio test ($p = 0.009$) and score (log-rank) test ($p = 0.020$).

Fig 2 presents a forest plot of multivariate Cox proportional hazard regression analysis, illustrating HRs and 95% CIs for all variables. Results of univariate analysis for all tested variables, including both significant and non-significant factors, are presented in Fig 2. Significant predictors of decreased survival included younger age and carotid artery abutment, while jugular bulb abutment and N stage approached significance.

This plot displays hazard ratios (HRs) with 95% confidence intervals for variables included in the multivariate Cox regression model assessing prognostic factors for survival in patients with EACC. A hazard ratio greater than 1 indicates an increased risk, while less than 1 indicates a protective effect. Statistically significant variables ($p < 0.050$) included age (protective) and carotid artery abutment. Error bars represent 95% confidence intervals. All variables examined in the univariate analysis are displayed irrespective of statistical significance.

## Discussion

The prognosis of EAC SCC varies considerably based on several factors, including tumor stage, surgical margins, lymph node involvement, perineural invasion, and adjuvant therapy [17]. Despite advancements in surgical and non-surgical treatments, long-term survival remains poor, particularly in advanced-stage disease [3]. In this study, advanced T-stage, nodal metastasis, and vascular abutment—particularly involving the carotid artery—were associated with worse OS, while older age showed a protective effect. These findings highlight the prognostic value of tumor stage and anatomical involvement.

Tumor staging remains the most significant prognostic factor in EAC SCC. The Modified Pittsburgh classification, which integrates radiologic and pathologic findings, remains the most widely used system and has been validated in several landmark studies [18,19]. In these reports, survival outcomes were primarily stratified by overall T stage, with T1–T2 tumors demonstrating significantly better survival than T3–T4 tumors [17]. Consistent with these findings, our results showed excellent outcomes in early-stage disease (100.0% 5-year OS for T1–T2) and markedly poorer survival in advanced stages (60.0% for T3 and 42.0% for T4). These findings further support the oncologic adequacy of surgery alone for early-stage EAC SCC when negative margins are achieved.

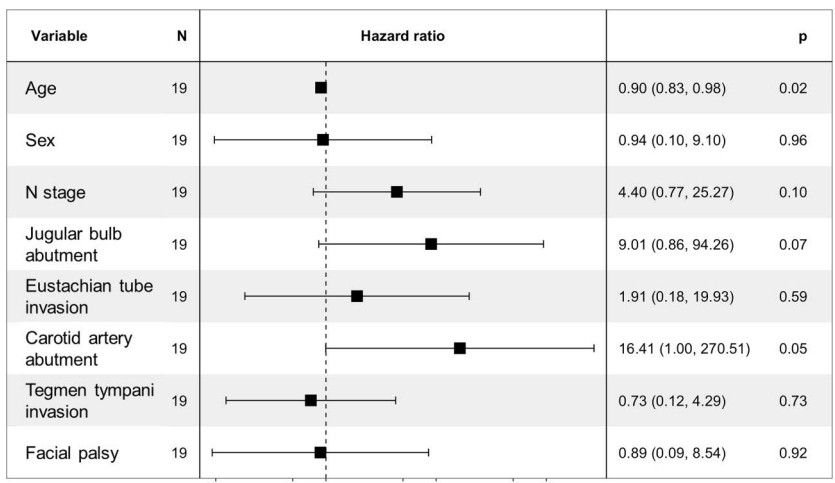

**Fig 2. Forest plot of multivariate cox proportional hazard regression analysis.**

Our study also provides one of the largest single-institution datasets of external auditory canal squamous cell carcinoma, including a substantial number of advanced-stage (T3–T4) cases collected over more than two decades. This large, homogeneous cohort enabled a detailed survival analysis within advanced disease, allowing refined evaluation of specific anatomical invasion sites—such as the TMJ, mastoid, and dura—and their prognostic implications. These findings extend the existing staging framework and offer new insight into the heterogeneity of advanced EAC SCC.

Achieving negative surgical margins is essential for curative resection, but this becomes challenging in advanced tumors with invasion of critical structures. As in previous studies, locally advanced disease was the strongest predictor of poor prognosis. Vascular abutment often precludes en bloc resection and was strongly associated with poorer outcomes in our cohort. Positive surgical margins were associated with a significantly higher recurrence rate and worse survival outcomes [3,20], emphasizing the need for complete tumor resection with adequate margins. In our cohort, initial surgical margin positivity closely reflected site-specific invasion of critical structures such as the carotid artery, jugular bulb, and Eustachian tube. Therefore, these anatomical invasion variables were used as surrogate markers for extent of disease rather than including margin status as an independent covariate to avoid multicollinearity in multivariate modeling. Importantly, no patient was left with gross residual tumor. Any macroscopic remaining tumor was removed by additional stepwise excision guided by intraoperative frozen-section confirmation until all accessible margins were cleared. Thus, initial margin positivity did not indicate inadequate oncologic resection. Perineural invasion in this cohort was clinically manifested almost exclusively as facial nerve involvement. Because isolated histopathologic perineural invasion was extremely rare and not statistically analyzable as a separate variable, facial nerve palsy was used as a clinical surrogate marker of perineural invasion in the prognostic analyses. Carotid artery abutment was an independent negative prognostic factor in our analysis, while jugular bulb abutment showed borderline significance. These results highlight the critical importance of early diagnosis before tumors invade essential anatomical structures and curative resection becomes unfeasible.

Recent evidence suggests that habitual ear picking and related chronic irritation may contribute to the development of EAC SCC, particularly in East Asian populations [21,22]. However, the causal direction remains unclear—repeated mechanical irritation could promote carcinogenesis, but such behavior may also arise secondary to early neoplastic changes or preexisting local discomfort. In our cohort, tumor laterality did not show a notable predominance (right 25 vs. left 31), and handedness data were unavailable. Therefore, although this behavioral hypothesis is intriguing, additional prospective studies are required to determine whether ear manipulation represents a true etiologic factor or merely an epiphenomenon of early disease.

A previous retrospective cohort study found that perineural invasion, and venous and lymphatic involvement, significantly worsened OS and DSS, supporting the role of aggressive multimodal therapy [23]. Although these histological features did not reach statistical significance in our cohort, nodal metastasis demonstrated a borderline association with poorer outcomes, suggesting a potential role for the selective use of adjuvant therapy in high-risk patients.

Facial nerve palsy is a known adverse prognostic factor. A prior study found that preoperative facial nerve involvement increased the risk of mortality substantially (HR for OS and DSS was 3.80 and 7.63, respectively) [24]. In our study, facial palsy was not an independent predictor of survival. However, five of the six surviving patients with T4 tumors had normal facial function, further supporting the notion that facial nerve involvement reflects aggressive disease and warrants multidisciplinary management.

Surgical resection remains the cornerstone of treatment for EAC SCC, particularly for advanced-stage tumors. Achieving clear surgical margins is often challenging due to the proximity of vital structures. STBR offers the highest chance of achieving negative margins but is associated with substantial morbidity. Preoperative imaging and meticulous surgical planning are essential. In our cohort, no cases involved direct invasion of the carotid artery or jugular bulb. However, when vascular abutment was present, tumors were excised with minimal margins, followed by cauterization of the surrounding tissues. Postoperative adjuvant RT or CRT was administered to ensure complete eradication of any residual disease. It should be emphasized that positive surgical margins in our analysis referred to the initial en bloc resection specimen

rather than the final surgical field status. In advanced T4 cases, residual tumor was commonly removed through additional stepwise excision guided by intraoperative frozen-section assessment to achieve clearance of all accessible margins. Therefore, initial margin positivity does not equate to incomplete oncologic treatment but reflects the technical complexity of en bloc removal in tumors abutting critical structures.

Regarding Case 13, which demonstrated synchronous lung metastasis at diagnosis; however, distant metastasis was not an absolute exclusion criterion for surgical intervention in our treatment strategy. This patient was considered to have oligometastatic disease and underwent surgical resection for loco-regional disease control in conjunction with systemic therapy for distant metastasis. Therefore, surgery was performed with therapeutic intent rather than palliative debulking. This approach reflects real-world multidisciplinary management of select advanced EAC SCC cases.

Regarding cases with preoperative imaging suggesting carotid artery (CA) involvement, it should be clarified that these represented tumor abutment to the vascular surface rather than true intramural or intraluminal invasion. None of the patients in our cohort demonstrated destructive infiltration of the CA wall requiring vascular sacrifice. In these cases, en bloc tumor resection was initially performed, followed by additional targeted excision and careful scraping of the adherent tumor tissue from the CA surface until macroscopic clearance was achieved. Thus, these procedures were undertaken with curative intent rather than palliative debulking. Although all patients with CA abutment ultimately succumbed to disease, this reflects the aggressive nature and extensive skull-base or intracranial involvement of advanced T4 tumors rather than technical futility of surgical intervention itself.

With respect to jugular fossa and jugular bulb involvement, we differentiated bony invasion of the jugular fossa from venous surface abutment of the jugular bulb based on both preoperative imaging and intraoperative findings. Jugular fossa involvement, characterized by skull-base bone destruction, can be reliably identified on preoperative CT and MRI. In contrast, jugular bulb "involvement" in our study referred to tumor abutment or adherence to the venous surface without evidence of intraluminal invasion or venous wall penetration. None of our cases demonstrated true jugular bulb invasion requiring venous sacrifice or resection. When jugular bulb abutment was encountered intraoperatively, additional careful surface excision and scraping were performed to achieve macroscopic clearance; in one case, partial surface reconstruction using absorbable hemostatic materials (e.g., Gelfoam) was required. Therefore, the term "no direct invasion of the jugular bulb" used in this manuscript indicates the absence of intraluminal venous invasion, which is consistent with our surgical management and pathological findings.

The role of adjuvant RT and CRT in improving survival outcomes remains controversial. However, in patients with locally advanced disease (T3–T4), postoperative RT is widely practiced even after negative-margin resection, with the primary goal of improving local disease control rather than overall survival [18,19]. Given the aggressive infiltrative behavior of EAC SCC and the difficulty in ensuring true histological clearance from surrounding critical structures, this multimodal approach is generally considered standard care rather than overtreatment. Adjuvant RT may improve local control without significantly affecting OS [25]. A 5-year LRC of 73.9% with adjuvant RT compared with 45.8% for definitive RT highlights its value in postoperative settings. CRT showed modest benefit in inoperable T2–3 tumors, although prior radiation exposure negatively impacted outcomes, highlighting the need for judicious patient selection [26]. Within the scope of EAC SCC, RT and CRT are generally reserved for adjuvant therapy following surgical resection. Chemotherapy regimens varied substantially across the 20-year study period, as many patients received treatment at local affiliated hospitals. Therefore, protocol heterogeneity limited the ability to compare systemic therapies directly or assess their independent prognostic impact. This highlights the challenges in establishing standardized multimodal treatment strategies for such a rare malignancy.

Balancing oncological control with functional preservation is central in managing advanced EAC SCC. En bloc resections such as LTBR or STBR maximize margin clearance but are associated with significant morbidity, including facial nerve paralysis, hearing loss, and cerebrospinal fluid leaks. Treatment decisions must weigh tumor stage, patient comorbidities, and anticipated quality of life. All surviving T4 patients in our study underwent STBR. In cases of vascular

abutment, tumors were resected with minimal margins, followed by cauterization and adjuvant therapy. Given the aggressive nature of advanced EACC, a multidisciplinary approach is essential. Future advancements in RT and molecular-targeted therapies may improve outcomes, but current strategies continue to rely heavily on early detection and complete surgical resection. Prospective studies and molecular profiling are warranted to establish standardized treatment protocols and guide personalized therapy.

## Conclusion

EACC, particularly in advanced stages, poses a formidable clinical challenge due to its complex anatomical location and aggressive behavior. Optimal management requires a multidisciplinary approach, with complete surgical resection and negative margins being critical for improving outcomes. Tumor stage, nodal status, and vascular abutment, particularly involving the carotid artery or jugular bulb, are key prognostic indicators. While early-stage tumors are associated with favorable outcomes, advanced-stage disease is marked by high recurrence and poor long-term survival. Given the challenges in achieving clear margins in anatomically constrained regions, early diagnosis remains essential to enable curative en bloc resection and improve prognosis.

## Supporting information

**S1 Dataset. Anonymized clinical and treatment data of patients with external auditory canal squamous cell carcinoma analyzed in this study.**
(XLSX)

## Author contributions

**Conceptualization:** Jong Woo Chung.

**Data curation:** Woo Seok Kang, Joong Ho Ahn, Hong Ju Park, Jong Woo Chung.

**Formal analysis:** Se Eun Yi.

**Methodology:** Byung Chul Kang, Se Eun Yi, Jong Woo Chung.

**Project administration:** Byung Chul Kang, Jong Woo Chung.

**Supervision:** Jong Woo Chung.

**Validation:** Jong Woo Chung.

**Visualization:** Byung Chul Kang, Se Eun Yi.

**Writing – original draft:** Byung Chul Kang, Se Eun Yi.

**Writing – review & editing:** Byung Chul Kang, Jong Woo Chung.

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
