## [Decision Letter · Decision Letter 0]

2 Oct 2025

Dear Dr. Chung,

We look forward to receiving your revised manuscript.

Kind regards,

Sethu Thakachy Subha, M.S

Academic Editor

PLOS ONE

Journal Requirements:

2. In the online submission form, you indicated that [The datasets generated and/or analysed during the current study are not publicly available due to patient privacy protections, but are available from the corresponding author upon reasonable request.].

Reviewers' comments:

Reviewer's Responses to Questions

**Comments to the Author**

1. Is the manuscript technically sound, and do the data support the conclusions?

Reviewer #1: Yes

Reviewer #2: Yes

Reviewer #3: Yes

2. Has the statistical analysis been performed appropriately and rigorously?

Reviewer #1: Yes

Reviewer #2: Yes

Reviewer #3: Yes

3. Have the authors made all data underlying the findings in their manuscript fully available?

Reviewer #1: Yes

Reviewer #2: No

Reviewer #3: No

4. Is the manuscript presented in an intelligible fashion and written in standard English?

Reviewer #1: Yes

Reviewer #2: Yes

Reviewer #3: Yes

Reviewer #1: In this manuscript entitled “Treatment Outcomes and Prognostic Factors in External Auditory Canal Squamous Cell Carcinoma”, written by Kang et al, the authors retrospectively reviewed clinical, pathological and survival data of 56 patients with External Auditory Canal Squamous Cell Carcinoma (EACC). The authors performed survival analysis as well as multivariate analysis. The statistical methods in this study are mostly appropriate and the authors discussed their results comparing the previous studies of EACC. Overall, the manuscript is in good quality. I have few comments below.

Figure 1A and 1B. Please include the table for "number at risk" under the Kaplan-Meier curves so that the readers easily understand the changes in the patient numbers tested. Also, censoring points should be indicated in the plots.

I strongly recommend the authors include T1 and T2 patients in the Kaplan Meier plots as well. Since the authors noted the survival rate for T1 and T2 patients was 100%, including these groups in the figure may indicate a rationale for the good quality of treatment the authors provided.

Figure 2. How did the authors divide the patients by age into “young” and ‘’old” for univariate and multivariate analyses? I could not find the cutoff for age in these analyses.

Recent clinical and translational research (Tsunoda et al, Laryngoscope Investig Otolaryngol. 2017, PMID: 28894818; Sato et al, Cancer Sci. 2020, PMID: 32500594) indicated that habitual ear picking and subsequent chronic inflammation could be a common risk for EACC (I assume that this would be particularly in eastern Asian population). Did the authors investigate potential risk factors and/or clinical backgrounds in these patients? The authors may discuss this point citing the publications above.

Reviewer #2: This paper is a retrospective review of 56 cases of squamous cell carcinoma of the external auditory canal. Focusing on squamous cell carcinoma alone is a strength of the paper. That early-stage tumors (T1 and T2) outnumbered advanced stage disease is an unusual feature of the study population. IRB approval was obtained for this study. Inclusion and exclusion criteria are outlined. The modified Pittsburgh classification was used for staging. The paper is well written.

There are significant deficiencies in this paper. The authors do not detail the adjuvant therapy given to this cohort. Under Methods, the authors mention that “Adjuvant radiotherapy was administered in cases with positive margins, perineural invasion, or advanced disease features,” but we are not told who had positive margins, etc. Under Surgical Methods and Survival Outcomes, the authors write that none of the T1 or T2 patients received radiotherapy, while all T3 and T4 patients received postoperative chemoradiotherapy. The chemotherapy agents are not mentioned. The radiotherapy doses are not mentioned.

The authors should provide more details regarding the entire cohort of patients. In Table 2, the authors provide details on T4 patients; this level of detail should be provided for all patients.

Suggestions, questions, and comments:

1. The Introduction is too long. It should be no longer than 3 paragraphs. The Introduction should contain the salient facts leading to the study, especially noting the controversies or innovations.

2. Surgical Methods and Survival Outcomes, it is unclear why the authors tabulated only the T4 cases. Such granular data for all study patients would be of interest to researchers worldwide. For this reviewer, comparing the outcomes of patients who underwent sleeve resection to those who underwent lateral temporal bone resection would be of significant value.

3. Table 2 does not mention any adjuvant treatment for this T4 tumors. Did any of these patients receive either chemotherapy or radiotherapy?

Reviewer #3: The article is potentially interesting, since it provides a detailed description of the survival outcomes of a homogeneous series of SCC of the external acoustic canal.

The most relevant strenght of this article resides in a clear description of the characteristics of the included subjects, and of the anatomical sites of invasion in the advanced-stage cases.

However, it should be more clearly explained what the results of this article add to the current sate of knowledge on this field.

Although Authors provided a detailed calculation of the mortality risk (both in univariate and multivariate setting) for each demographic and clinical variable (including a detailed breakdown of the prognostic role of different anatomical invasion sites), the presented data seem to confirm what was already known from the literature (in example, doi: 10.1002/hed.2643; doi: 10.1177/019459989110500102; doi: 10.1055/s-0030-1253575 etc.).

For this reason, I suggest to expand the discussion, developing more in-depth the actual meaning of this results with reference to the existing literature (including the existing different staging methods and comprehensive prognostic scores).

Minor concerns:

- does "CA" mean carotid artery involvement in Tab. 2? Please, explain it in the abbreviations.

- maybe including the KM graphs also for T1 and T2 cases may improve Fig. 1A and B

**Do you want your identity to be public for this peer review?** For information about this choice, including consent withdrawal, please see our Privacy Policy

Reviewer #1: No

Reviewer #2: No

Reviewer #3: No

---

## [Author Response · Author response to Decision Letter 1]

5 Nov 2025

Dear Reviewers,

We are pleased to submit the revised version of our manuscript,

“Treatment Outcomes and Prognostic Factors in External Auditory Canal Squamous Cell Carcinoma.”

All reviewer comments have been carefully addressed, and corresponding changes are highlighted in yellow in the revised manuscript.

We appreciate your consideration of our revision.

Please refer to the uploaded file titled “Response to Reviewers,” which provides our detailed, point-by-point responses.

---

## [Decision Letter · Decision Letter 1]

20 Nov 2025

Dear Dr. Chung,

Thank you for submitting your manuscript to PLOS ONE. After careful consideration, we feel that it has merit but does not fully meet PLOS ONE’s publication criteria as it currently stands. Therefore, we invite you to submit a revised version of the manuscript that addresses the points raised during the review process.

We look forward to receiving your revised manuscript.

Kind regards,

Sethu Thakachy Subha, M.S

Academic Editor

PLOS ONE

**Journal Requirements:**

Reviewers' comments:

Reviewer's Responses to Questions

**Comments to the Author**

Reviewer #1: All comments have been addressed

Reviewer #4: All comments have been addressed

Reviewer #5: All comments have been addressed

2. Is the manuscript technically sound, and do the data support the conclusions?

Reviewer #1: Yes

Reviewer #4: Partly

Reviewer #5: Partly

3. Has the statistical analysis been performed appropriately and rigorously?

Reviewer #1: Yes

Reviewer #4: Yes

Reviewer #5: No

4. Have the authors made all data underlying the findings in their manuscript fully available?

Reviewer #1: Yes

Reviewer #4: Yes

Reviewer #5: Yes

5. Is the manuscript presented in an intelligible fashion and written in standard English?

Reviewer #1: Yes

Reviewer #4: Yes

Reviewer #5: Yes

Reviewer #1: Thank you for responding to my questions. All of my questions have been solved properly, and the quality of the manuscript has been significantly improved.

Reviewer #4: Jong Woo Chung et al. present a retrospective study of 56 cases of external auditory canal squamous cell carcinoma (EAC-SCC) treated with surgical intervention. The relatively large number of cases and the follow-up data after treatment make this study a potentially valuable contribution. As a consolidated case series from China, the report has significance; however, it unfortunately lacks novelty. Moreover, essential information typically expected in a case series is not sufficiently provided.

Minor Points:

#1. The number of significant digits used in the manuscript is inconsistent and should be standardized.

#2. It should be clearly stated whether the TNM staging presented is clinical or pathological.

Major Points:

#1. Whether the surgeries were performed via en bloc resection or piecemeal resection is critically important for oncologic discussion and must be described explicitly.

#2. The manuscript states: “For T3 and T4 lesions, all patients underwent postoperative adjuvant chemoradiotherapy (CRT).” However, administering postoperative radiation therapy in all advanced cases, even when surgical margins are negative, may constitute overtreatment from an oncologic perspective.

#3. Data on postoperative margin status are lacking. Margin assessment is essential information. It is problematic to engage in discussion regarding oncologic outcomes without presenting data on surgical margins. At a minimum, margin status should be included in the case profile of T4 patients (Table 2).

#4. The rationale for performing surgery on cases with preoperative imaging suggesting carotid artery (CA) involvement should be clarified. According to existing literature, complete resection with negative margins is theoretically impossible in such cases, and curative treatment is considered extremely difficult even with adjuvant therapy. Given that all such patients in this series ultimately died, the justification for surgical intervention in these cases should be discussed in detail.

#5. The manuscript distinguishes between jugular fossa and jugular bulb invasion in preoperative imaging. However, is it truly feasible to differentiate these two sites accurately based on imaging alone? The basis for this distinction should be clarified. If jugular bulb invasion was diagnosed preoperatively, resection of the involved vein should have been planned in order to achieve negative margins. In practice, direct visual confirmation of venous invasion is not possible intraoperatively, as the tumor would already be exposed at that point. Despite this, the Discussion section states: “no cases involved direct invasion of the carotid artery or jugular bulb,” which appears contradictory if the preoperative strategy was to achieve negative margins.

Reviewer #5: This paper investigates prognostic factors for external auditory canal carcinoma, focusing mainly on the impact of invasion sites in T4 cases. Although similar studies have been published previously and no new prognostic factors were identified, the paper holds value as a case series.

There are several points of concern that require revision, which I have listed below.

1) In surgical procedure section, line 8. Does this mean that neck dissection was performed in all T1 and T2 cases, but no lymph node metastasis was found in any of them? At the same time, it is also necessary to describe how the parotid gland was managed. Additionally, please indicate whether neck dissection and parotidectomy were performed in T3 and T4 cases.

2) In surgical methods and survival outcomes section, line2. T2 cases involve bone invasion, and typically cannot be treated with sleeve resection. Please verify this.

3) In surgical methods and survival outcomes section, line4. It would be better to clearly state that all T1 and T2 cases had negative surgical margins. In addition, a description of the surgical margins for T3 and T4 cases is also necessary.

4) It is stated that extended LTBR was performed in six T4 cases; however, Table 2 lists only three such cases. Please correct this inconsistency. In addition, Table 2 includes only 19 out of the 21 T4 cases. Furthermore, in Case 13, distant metastasis is reported. Since surgery would not be considered a curative treatment in such a case, this case should be excluded.

5) In surgical methods and survival outcomes section, line6. It is stated that all cases received postoperative CRT, but this contradicts later descriptions. This requires correction.

6) In adjuvant therapeutic patterns and results section, line4. The authors stated the involvement of multiple institutions, but I understand this study to be based on cases from a single institution. Could you clarify what is meant by this?

7) In adjuvant therapeutic patterns and results section, line1-2. It is written that two patients did not receive radiotherapy, and I assume this refers to two out of the 26 T3 and T4 cases.

8) I consider it an extremely favorable result that there were no recurrences in T1 and T2 cases treated with surgery alone, without postoperative radiotherapy. Although it is noted that the survival rate was 100%, please also provide information regarding local or regional recurrence, as well as any distant metastases.

9) In an univariate analysis, only the items that showed significant differences are reported. However, all items examined, including those with non-significant results, should also be presented. Regarding the univariate analysis of age, it is unclear what comparisons were made—for example, whether it was between patients aged 65 and older versus those under 65. This should be clearly specified. Furthermore, although a multivariate analysis was subsequently performed, it is not clear how the variables included in the multivariate model were selected.

10) The authors cite previous studies indicating that surgical margins and perineural invasion are poor prognostic factors. However, these factors are not included in the prognostic analyses of the current study. Please clarify the reason for their exclusion, and it would be desirable to include these factors in the analysis.

11) In cases with carotid artery invasion, achieving complete resection is generally difficult unless the carotid artery is sacrificed. However, in the present study, five such cases with carotid artery involvement are included. Since these cases are inherently challenging to treat with curative surgery, they would be considered to meet the exclusion criteria.

12) In the discussion, the authors emphasize the importance of achieving negative surgical margins and en bloc resection. However, this paper does not provide any data on margin status or whether en bloc resection was performed, and merely cites previous studies. At the very least, these data should be presented, and the study should demonstrate that these factors are indeed important prognostic indicators in the current cohort.

**Do you want your identity to be public for this peer review?** For information about this choice, including consent withdrawal, please see our Privacy Policy

Reviewer #1: No

Reviewer #4: No

Reviewer #5: **Yes: ** Hirotaka Shinomiya

---

## [Author Response · Author response to Decision Letter 2]

11 Dec 2025

In this second revision, we have carefully addressed all remaining comments from Reviewer #4 and Reviewer #5. The manuscript has been updated accordingly, and all changes are shown in track changes. A detailed point-by-point response has been submitted separately.

Revisions include:

1. Clarification of the analyzed T4 cohort and corresponding updates to Table 2;

2. Correction regarding adjuvant therapy non-completion (treatment refusal rather than ineligibility);

3. Added details on parotid gland management, neck dissection, and surgical margins; and

4. Updates to ensure consistency in the Results, Methods, and Discussion sections.

We believe the revised manuscript now fully addresses the reviewers’ concerns and is improved accordingly.

---

## [Editor Report · Decision Letter 2]

14 Dec 2025

Treatment Outcomes and Prognostic Factors in External Auditory Canal Squamous Cell Carcinoma

PONE-D-25-44347R2

Dear Dr. 

We’re pleased to inform you that your manuscript has been judged scientifically suitable for publication and will be formally accepted for publication once it meets all outstanding technical requirements.

Kind regards,

Sethu Thakachy Subha, M.S

Academic Editor

PLOS One
---

## [Editor Report · Acceptance letter]

PONE-D-25-44347R2

PLOS One

Dear Dr. Chung,

I'm pleased to inform you that your manuscript has been deemed suitable for publication in PLOS One. Congratulations! Your manuscript is now being handed over to our production team.

Kind regards,

on behalf of

Dr. Sethu Thakachy Subha

Academic Editor

PLOS One